# Respiratory and Otolaryngology Symptoms Following the 2019 Spring Floods in Quebec

**DOI:** 10.3390/ijerph191811738

**Published:** 2022-09-17

**Authors:** Elsa Landaverde, Mélissa Généreux, Danielle Maltais, Philippe Gachon

**Affiliations:** 1Département des Sciences de la Santé Communautaire, Université de Sherbrooke, 3001, 12e Avenue Nord, Sherbrooke, QC J1H 5N4, Canada; 2Département des Sciences Humaines et Sociales, Université du Québec à Chicoutimi, 555 Boulevard de l’Université, Ville de Saguenay, QC G7H21, Canada; 3Département de Geographie et Centre ESCER (Étude et Simulation du Climat à l’Échelle Régionale), Université du Québec à Montréal, 201, Avenue du Président-Kennedy, Montréal, QC H2X 3Y7, Canada

**Keywords:** floods, respiratory impacts, ENT symptoms, respiratory symptoms, primary stressors, secondary stressors

## Abstract

Background: Although floods may have important respiratory health impacts, few studies have examined this issue. This study aims to document the long-term impacts of the spring floods of 2019 in Quebec by (1) describing the population affected by the floods; (2) assessing the impacts on the respiratory system according to levels of exposure; and (3) determining the association between stressors and respiratory health. Methods: A population health survey was carried out across the six most affected regions 8–10 months post-floods. Data were collected on self-reported otolaryngology (ENT) and respiratory symptoms, along with primary and secondary stressors. Three levels of exposure were examined: flooded, disrupted and unaffected. Results: One in ten respondents declared being flooded and 31.4% being disrupted by the floods. Flooded and disrupted participants reported significantly more ENT symptoms (adjusted odds ratio (aOR): 3.18; 95% CI: 2.45–4.14; aOR: 1.76; 95% CI: 1.45–2.14) and respiratory symptoms (aOR: 3.41; 95% CI: 2.45–4.75; aOR: 1.45; 95% CI: 1.10–1.91) than the unaffected participants. All primary stressors and certain secondary stressors assessed were significantly associated with both ENT and respiratory symptoms, but no “dose–response” gradient could be observed. Conclusion: This study highlights the long-term adverse effects of flood exposure on respiratory health.

## 1. Introduction

Each year, the health of millions of individuals around the world is threatened by complex natural and human events, including conflicts and disasters. These events can be classified as natural (including extreme events such as storms, earthquakes and floods), man-made (including technological and industrial accidents, terrorism and war) or a combination of these two types [1]. Although each disaster presents unique characteristics, they all generate important immediate and long-term impacts on individuals and communities while also disrupting social systems and community functions, making recovery difficult [2]. The threat of natural disasters has always been present, but since the 1960s, the frequency of natural disasters related to hydrometeorological hazards has increased nearly three times to reach the level observed today [2,3,4]. In the province of Quebec (Canada), the severity and recurrence of floods seem to have increased in recent years over its eastern and maritime regions [5]. Climate change results in more frequent and intense precipitation, which, when combined with the rapid melting of the snow cover, can increase the risk of flooding [6,7]. However, the effects of climate change on flooding remain uncertain, as projections vary according to the region due to the complexity of factors involved [8]. Despite these uncertainties, many man-made factors, such as human exposure and vulnerability, infrastructures, shoreline modification and water management, greatly contribute to their incidence and severity [9].

Unlike the impacts of flooding on infrastructures, public services and the economy, which are widely recognized, their social and health impacts are less studied. Indeed, according to a systemic mapping by Zhong et al. (2018), only a small number of studies investigated physical, social or behavioural impacts following floods, in the short, medium and long term [10]. While the death rate related to flooding events is minor in Canada, these events can generate an important health burden on affected individuals [11,12]. These health problems can occur during or immediately after floods, notably with increased care seeking due to diarrhea, skin infections and acute respiratory infections [13,14], but can also have long-term sequelae, as individuals having been exposed to floods are more likely to describe their health as deteriorated two and three years after these events. They often report the appearance or exacerbation of certain physical health problems [15]. According to a quantitative meta-analysis, flood victims are more likely to be affected by respiratory issues such as respiratory infections and bronchitis in the weeks or months following an exposure. The respiratory infections category used in this analysis was broad, not allowing us to distinguish between types of respiratory symptoms [16]. Another study conducted 6 months after Hurricanes Katrina and Rita found that respiratory symptoms were positively associated with exposure to water-damaged homes [17]. This study did not, however, consider the level of exposure or quantify the water damage in the homes. Then, in the Netherlands, a study found that pluvial floodwater contact was significantly associated with acute respiratory infection, and there were additional risk factors, including skin contact with floodwater and performing post-flooding cleaning operations [18]. In Canada, many population subgroups, including lower socio-economic-level individuals, the young or old, ethnic minorities, non-insured individuals and individuals with pre-flood morbidities, are particularly vulnerable to these effects [19]. Findings suggest that these problems are exacerbated by the presence of visible mould in flooded homes, damp environments, water infiltration and humidity [16,19,20,21]. This suggests that individuals more heavily affected by floods may be more at risk for respiratory issues. Such environmental conditions have also been linked to allergic rhinitis and asthma [13,22]. A causal association has been found between dampness and exacerbation of asthma in children and association found in adults [23]. Using asthma as an indicator may, however, cause under-representation of symptoms, as only moderate to severe cases are usually recorded, as they lead to illness-related school absences and hospitalization. A meta-analysis by Jaakkola et al. (2013) reported that the largest risk factor for rhinitis was the presence of mould odour; however, visible mould and exposure to dampness were also related to increased risk of symptoms [24]. Other flood-related exposures, such as reconstruction exposure, skin contact with floodwater and performing post-flooding cleaning operations, also significantly increased the odds of presenting respiratory symptoms [17,20,24,25]. Although some international studies have identified respiratory impacts as an important issue following flooding events, findings are difficult to compare, as there does not seem to be a consensus surrounding the type of scale used to measure respiratory problems related to water damage. Outcomes ranged from lower respiratory symptoms in the 30 days preceding the survey [20], the presence of acute respiratory infection [18], a dichotomic respiratory health problem variable including symptoms of asthma, coughing or wheezing [21] and self-reported respiratory diseases [26], to the use of a “symptom score” for each participant by attributing a severity ranking to reported lower and upper respiratory symptoms [17]. In addition, some studies evaluated these impacts shortly following the event [18,21], whereas others evaluated respiratory symptoms 6 months to 1 year later [17,20].

Impacts of the disaster can be exacerbated by primary and secondary stressors [27]. Primary stressors are defined as direct results of a disaster that can occur during or immediately after the event. In the case of floods, for example, injuries or damage to dwellings caused by water [28]. On the other hand, secondary stressors have an indirect relationship with the event and arise during the recovery period [29]. They include consequences such as economic hardships, insurance complications, relocation, delays related to the cleaning or renovation of a flooded home and lack of social support received to cope with the difficulties experienced as a result a flood exposure [29]. Secondary stressors, strongly influenced by the interventions put in place by the government, may contribute to prolonged distress and complex health impacts following an event, as flood victims are dependent on external factors in order to adequately recover [30]. However, the relationship between flood-related respiratory impacts and secondary stressors remains unclear and is rarely presented in the current literature. This issue highlights the need to better understand the respective roles of primary and secondary stressors in respiratory health impacts of floods, in order to orient authorities’ efforts in the aftermaths of floods [27].

Evidence suggests that floods may have important respiratory health impacts on affected populations; however, few studies have examined these impacts in the long term. Quebec has been affected by various floods in recent years, affecting thousands of individuals in the province. A 57-year water level record resulting from simultaneous rapid ice cover melting and regular occurrences of intense precipitation in the spring of 2019 caused major flooding for the second time in three years [31]. In the context of these important spring floods, a particular event occurred in the city of Sainte-Marthe-sur-le-Lac in the Laurentians: the break of a dike whose role was to contain the waters of Lac des Deux-Montagnes [32]. The break caused the city to be filled with water, flooding 700 to 800 residences and the evacuation of one third of its citizens. In order to effectively prevent, prepare, intervene and recover from future events, a risk reduction approach must be favoured. This approach requires adequate recognition and understanding of the impacts of floods, as well as the context in which the floods occur in terms of vulnerability and exposure factors [33,34]. This strengthens the need to adequately understand the impacts in a Canadian context, even though flood-related respiratory impacts have been studied at the international level. In addition, few studies have explored the long-term effects while assessing various levels of flood exposure. The secondary stressors of floods have also rarely been analyzed in relation to their impacts on respiratory health. In this context, a study was put into place to document the state of health and vulnerabilities following the spring floods of 2019 in Quebec. This paper presents the quantitative results of the study focused on the respiratory impacts following the events. The objectives of the present article are to (1) describe the characteristics of the population affected by the floods; (2) assess the impacts on the respiratory system of spring flooding according to the different levels of exposure; and (3) determine the association between primary and secondary stressors and respiratory health following flooding.

## 2. Materials and Methods

### 2.1. Design

This investigation is rooted in the context of a larger mixed-method approach project led by an interdisciplinary and inter-university team joined through the Quebec Intersectorial Flood Network (RIISQ). This study is funded under the Québec government 2013–2020 Climate Change Action Plan. The quantitative section of the research consists of a population health survey undergone eight to ten months after the 2019 spring flood in Quebec across the six most affected socio-sanitary regions (Laurentides, Laval, Mauricie–Centre-du-Québec, Montérégie, Montréal, and Outaouais). The survey, conducted from December 2019 to February 2020, was carried out by telephone or by an online questionnaire under the supervision a professional survey firm.

### 2.2. Recruitment and Participants

In the six most affected regions, all areas (i.e., six-digit postal codes) in which at least one person’s residence or workplace was flooded during the 2019 spring flood were identified (*n* = 925 flooded areas). This information was used to compile all households located in these areas for which residential telephone numbers were available (*n* = 92,450 households). This allowed the polling firm to then randomly select phone numbers from this data bank and administer the survey via phone to those who accepted. Only one respondent per household, aged 18 or over, could answer the questionnaire. As a voluntary response sampling technique was used, additional steps were taken to ensure the adequate representation of hard-to-reach groups through residential phone lines, such as young adults. Sixteen thousand five hundred randomly selected households received a letter informing them that they would be contacted to respond to a survey and were provided the option of responding online. The recruitment procedure is further depicted in Figure 1. Despite these additional recruitment efforts, an overall response rate of 15.3% was obtained. A final sample size of 3437 households in flooded areas completed the questionnaire by telephone (*n* = 3138) or on the web (*n* = 299). Of this sample, 587 households resided in the municipality of Sainte-Marthe-sur-le-Lac located in the Laurentians. This oversampling was carried out in order to better understand the specific health issues among the citizens of this municipality, as the spring floods of 2019 in this municipality were aggravated by a particular technical incident (i.e., breaking of a dike). 

### 2.3. Measurement

The questionnaire (available in French and in English) lasted approximately 20–25 minutes and included 65 close-ended questions developed by the research team based on questionnaires from similar studies or validated measurement scales (See Appendix A for full questionnaire). Questions focused on the experience of flooding; primary stressors and secondary stressors; the physical (injuries, respiratory symptoms or illnesses, etc.) and psychological health (perceived mental health, symptoms of post-traumatic stress, anxiety or mood disorders, etc.); and personal characteristics (gender, age, level of education, income, marital status, main occupation, smoking status, etc.) of the respondents.

### 2.4. Outcomes

Respiratory health was examined in terms of self-reported symptoms. The frequencies of otolaryngology (ENT) symptoms and respiratory symptoms, other than episodes of flu, colds or seasonal allergies, were examined. Respondents were asked to estimate the frequencies of 11 ENT symptoms (red/burning/itchy eyes, stuffy nose, runny nose, sneezing, dry nose, nosebleed, pressure or pain in the ear, sore throat, dry throat, secretion in the throat and watery eyes) and 5 respiratory symptoms (cough, sputum, wheezing, shortness of breath, chest tightness) during the last six months preceding the survey. Specific frequent ENT (or specific frequent respiratory) symptoms were defined as the presence of at least two ENT (or respiratory) symptoms reported at a frequency of at least 2–3 days per week, that had been improved or stayed the same when outside of the primary residence. This measurement scale was inspired by a scale widely used by public health authorities in the province of Quebec. It is employed in the clinical setting for the identification of probable cases of mould-related respiratory illness linked to building sanitation. In order to simplify the text, the self-reported symptoms will be referred to as ENT and respiratory symptoms in this article.

### 2.5. Exposure to Floods and Stressors

In order to observe the effects of floods on different respiratory health outcomes, our definition of exposure was largely based on that of a national flood and health survey conducted in England [35]. Thus, three levels of exposure were examined among people living in flooded areas, namely (1) having been flooded (direct exposure), (2) having experienced flood-related disruptions (indirect exposure), and (3) not having been affected by the floods. A person was considered flooded (directly exposed to flooding) if he/she reported having at least one liveable room flooded. An individual was considered disrupted (exposed indirectly) if he/she did not have a flooded liveable room, but reported at least one of the following disruptions during the floods: evacuation; interruption of home services; difficulty accessing community services; flooded non-liveable areas. Unaffected people were those who had no flooded rooms and who did not experience flood-related disturbances. Questions concerning exposure to primary and secondary stressors were also asked in the survey. Primary stressors (occurring during or immediately after the floods) included water levels in home, extent of material losses and recurrence of floods. Secondary stressors (occurring during the recovery period) included negative perception about concrete or moral help received, lack of financial help received to meet costs, lack of insurance covering floods, use of bank loans to meet expenses and inability to reuse all rooms. These stressors were only assessed in participants who were flooded or disrupted during the 2019 floods.

### 2.6. Analysis

Chi-square tests and Z-tests were performed with SPSS software version 26 to compare respiratory health outcomes across the three levels of exposure (flooded, disrupted and unaffected) in order to identify if the observed impacts significantly differed between each group. These tests were also performed to identify the differences in respiratory and ENT symptoms according to socio-demographic variables. Bivariate and multivariate logistic regressions were carried out to assess the crude and adjusted associations between primary and secondary stressors, and respiratory health outcomes.

## 3. Results

### 3.1. Respondent Characteristics

The 3437 participants originated from the six following regions: Laurentides (*n* = 1423), Outaouais (*n* = 794), Laval (*n* = 134), Mauricie-Centre du Québec (*n* = 643), Montérégie (*n* = 326) and Montréal (*n* = 117). Almost half of the respondents (46%) declared themselves to be victims of the 2019 flood (either flooded or disrupted), including 10.2% being flooded and 31.4% being disrupted by the floods. However, respondents in Sainte-Marthe-sur-le-Lac reported having been much more affected by the floods (30.3% flooded and 57.1% disrupted). Flood exposure according to each region studied is depicted in Table 1.

Table 2 presents the prevalence of primary and secondary stressors in the flooded areas under study. The Sainte-Marthe-sur-le-Lac municipality was analyzed separately from the other regions, given the particular context of their floods in 2019. Both primary and secondary stressors were found to be more frequent in this particular area. For example, 24.5% of respondents in Sainte-Marthe-sur-le-Lac reported water level of at least 30 cm (at the first floor), and 65.5% had received an amount for expenses equivalent to about half or less of the damage costs, as opposed to much lower prevalence in the other regions (3.4% and 53.2% respectively, *p* ≤ 0.001 and *p* ≤ 0.02). Furthermore, significantly more individuals required a bank loan to meet expenses (*p* ≤ 0.001), and more than two times as many remained unable to reuse of all rooms normally after the floods compared to individuals in the other regions (*p* ≤ 0.001). Affected households in the municipality of Sainte-Marthe-sur-le-Lac seem, however, to have felt better socially supported during recovery, as 25.7% reported having received less moral or concrete help than expected (vs. 35.9% elsewhere, *p* ≤ 0.005). 

### 3.2. Respiratory Health Impacts

When observing respiratory health impacts according to flood exposure, as seen in Table 3, the flooded participants reported significantly more frequent ENT and respiratory symptoms (34.7% and 19.8%, respectively), compared to the disrupted (21.7% and 8.8%, respectively) and the unaffected (14.6% and 6.9%, respectively). When observing specifically the ENT symptoms for all affected regions, a significant gradient was observed for ENT symptoms according to the level of exposure to floods, whereas no difference was found between the disrupted and the unaffected for respiratory symptoms. Despite greater primary and secondary stressors observed in Sainte-Marthe-sur-le-Lac, as presented in the previous table, flooded participants from this area reported similar frequencies of ENT and respiratory symptoms to those living in other regions.

The respiratory health outcomes were then analyzed according to personal characteristics and presented in Table 4. Only a few factors seemed to be associated with more frequent ENT and respiratory symptoms in flooded individuals—notably, smoking. However, these differences were also observed in the disrupted and the unaffected groups.

### 3.3. Associated Factors

Crude and adjusted odds ratios generated using bivariate and multivariate logistic regressions testing the correlations between primary and secondary stressors and respiratory health outcomes are presented in Table 5 and Table 6, respectively. All the primary stressors assessed were associated with both ENT and respiratory symptoms. However, these impacts did not follow a clear gradient according to the severity of the stressors, with the exception of the level of exposure. For ENT symptoms, the odds ratio (aOR) was 3.18 (95% confidence interval (CI) 2.45–4.14) for the flooded group and 1.76 (95% CI 1.45–2.14) for the disrupted group (reference: unaffected group), after adjustment for personal characteristics. For respiratory symptoms, the odds ratio (aOR) was 3.41 (95% CI 2.45–4.75) for the flooded group and 1.45 (95% CI 1.10–1.91) for the disrupted group.

Compared to primary stressors, secondary stressors were less strongly associated with respiratory health outcomes. Nonetheless, ENT symptoms were more likely to be observed in those who received less concrete or moral help than expected (aOR 1.87; 95% CI 1.11–3.13), who received a financial sum to cover damages that were less than half the costs (aOR 2.44; 95% CI 1.51–3.93), who took a bank loan to meet expenses (aOR 2.54; CI 1.66–3.89) and who could not reuse all rooms normally a year after the flood (aOR 2.77; CI 1.96–3.91). These stressors were also significantly associated with respiratory symptoms, with the exception of the amount of concrete or moral help received. For both ENT and respiratory symptoms, not having insurance covering floods did not significantly increase the risk of reporting symptoms. Similarly to primary stressors, crude and adjusted odd ratios were similar, demonstrating the small impact of personal characteristics as confounders.

## 4. Discussion

This study highlights the long-term adverse effects of flood exposure on the respiratory health of floods victims. Eight to ten months following the event, flooded individuals were about three times more likely to report ENT and respiratory symptoms than the unaffected living in the same areas, independent of personal characteristics. Although few socio-demographic factors seemed to have influenced the respiratory health in flooded individuals, they do appear to have a significant impact in those who have been disrupted or unaffected by the floods. These findings suggest that a direct exposure to floods may generate symptoms regardless of one’s characteristics, but also that these factors influence one’s vulnerability with a lesser exposure. All primary stressors studied (e.g., height of water in the home) were significantly linked to the development of ENT and respiratory symptoms. Even though greater exposure to these stressors was associated with higher ORs, some overlap in confidence intervals between levels (e.g., less than 30 cm, 30–100 cm, more than 100 cm) prevented us from concluding on clear “dose–response” gradients. The results of the analyses also demonstrate that no significant difference was noted between the region of Sainte-Marthe-sur-le-Lac and the rest of the affected regions in terms of prevalence of respiratory issues. This suggests that flood victims had similar respiratory health impacts following the 2019 floods, despite the participants from Sainte-Marthe-sur-le-Lac reporting being more severely affected by the floods, including higher levels of water and a larger extent of material losses. This echoes the results of the multivariate analyzes discussed previously. The impacts of secondary stressors on respiratory health were also explored in this study. ENT symptoms were more likely to be observed in those who received less concrete or moral help than expected, who received an insufficient financial sum to cover damages, who used a bank loan to meet expenses and who were unable to reuse of all rooms normally a year after the flood. Most of these stressors were also significantly associated with respiratory symptoms. Secondary stressors have been shown to greatly influence negative mental health impacts following a disaster, but prior to this study, few studies examined their impacts on the respiratory health of disaster victims [28,29]. This study also puts light on the magnitude of these stressors: 32% of participants expressed receiving a less than expected amount of concrete or moral help, and 29% reported receiving a financial sum that covered less than half the costs of their flood-related expenses.

Overall, our findings are consistent with those established previously. Like the present study, others have identified exposure to a flooding event as significantly associated with increased odds of reporting respiratory symptoms, even several months after the event [17]. Similarly, a study in the Tri-state metropolitan area following Hurricane Sandy found that over one third of the participants reported post-Sandy lower respiratory symptoms. Participants exposed to mould and damp environments had about twice the odds of reporting these symptoms compared to unaffected participants [20]. Other studies have also found secondary stressors to be largely prevalent following a flood event. A cross-sectional survey following the 2011 Queensland floods found even higher prevalence of secondary stressors: 44% of respondents reported not receiving adequate financial compensation from the government or their insurance to cover the damage, and 63% reported not receiving adequate community support [36]. Some of the results obtained in this study did, however, differ from the current literature. One study found that the magnitude of exposure acted as a risk factor for acute respiratory infections [18], whereas our study did not find a significant “dose–response” gradient with more severe stressors. However, the researchers noted that the self-reporting of water height could have been a potential bias [18]. Although this bias may also be present in our study, the use of two different exposure variables (water levels and estimated material losses) with an association could have strengthened our findings. It is important to note that comparing results is difficult due to the lack of a standardized scale to measure respiratory symptoms caused by water-related damage. This study emphasizes the importance of creating a standardized tool in order to adequately measure and compare impacts of floods on respiratory health.

These findings are eye-opening in the context of recovery interventions, as the results highlight the importance of appropriate cleaning, disinfection and renovation post-flood, regardless of the amount of water that entered the home. The results reinforce current knowledge about health consequences following floods, as damp environments harbouring mould have been linked to respiratory symptoms such as wheezing and coughing, and can cause more serious issues in individuals with underlying conditions such as asthma [20]. Mould and bacteria multiply in environments that sustain sufficient moisture levels for a certain period of time, increasing the risk of allergic reactions and respiratory problems in residents due to aerosolization of spores, toxins and other harmful byproducts [37]. The relationship between respiratory symptoms and secondary stressors is less direct, but it is hypothesized that the link may also stem from their impact on mould proliferation. Those who expressed receiving less concrete or moral help than expected may have performed cleaning or restoration less rapidly or less efficiently, resulting in inadequate removal of mould. They may have also performed the cleaning activities without personal protective equipment, further exposing themselves to mould and other toxins [38]. These findings are consistent with those of a longitudinal study evaluating the post-Hurricane-Katrina onset of respiratory issues. The prevalence rate ratios for sinus symptoms, fever and cough were significantly elevated for those who participated in restoration work [25]. Furthermore, those who had contact with flood water in addition to participating in cleaning and reconstruction operations (either inside or outside of a residence) were even more at risk of reporting these symptoms [18]. Many victims may not have the financial means to carry out the appropriate interventions, leading to inappropriate cleaning and mould removal. In addition, mould contamination often generates very expensive remediation efforts, which could explain why those facing larger costs would also be more at risk of developing respiratory issues [16]. Finally, the inability to reuse of all rooms normally translates to delays in cleaning activities or restoration, which could increase the risk of mould exposure, as cleaning and drying should be performed in the 48 h following water exposure to eliminate the growth of mould [39].

A limitation of this study is the low response rate of the survey, despite the extensive efforts made to achieve a high response rate. Not only did the survey contain sensitive topics for floods victims, but it was also carried out amid disaster recovery, with many people still in the process of obtaining financial assistance, rebuilding their homes or relocating. It is also possible that only those least affected by the floods or the people least satisfied with the management of the floods agreed to answer the survey, wishing to share their frustration. Another limitation lies in the inability of accessing cellphone numbers from postal codes. This may explain the low representation of young adults compared to older people in this study. The web alternative to responding to the survey also yielded low response rates. This confirms that a non-personalized letter would not have been a good alternative strategy for reaching participants. Finally, this study did not allow us to know if the respondents had respiratory problems before the 2019 floods. Since several of the respondents were also flooded in 2017, there is a possibility that their respiratory problems were caused by these floods. Despite these limitations, this study remains one of the largest population-based studies documenting the impacts of floods on the health of individuals who were directly or indirectly exposed to this event. A strength of our methodology was evaluating the respiratory impacts through the measurement of symptoms rather than using data on diagnoses. This method can be more inclusive, as diagnostic testing is not accessible for everyone, especially following a disaster. Then, this study considered several primary and secondary stressors, along with reviewing long-term effects—a rarity in the current literature. Unfortunately, the cross-sectional nature of this study did not allow us to evaluate the evolution of these impacts over time or to establish a causal link. Survey data are relatively static, hindering the possibility of observing the evolution of the impacts of floods [40]. This research project does, however, contain a second phase, thanks to participants who agreed to be recontacted for another survey a few months after this study. The analysis of the data collected during this second phase will allow a better understanding of the prolonged effects of the floods in the future, and this is currently under evaluation.

## 5. Conclusions

This study demonstrated the importance of taking primary and secondary stressors into account when studying the impacts that flooding can have on the respiratory health of people exposed to this type of event. Unfortunately, long-term recovery is the ignored phase of emergency management, and existing knowledge about this phase is seriously lacking [41]. Low levels of interest are also given to disaster recovery, most probably because the emergency response, compared to recovery, requires immediate attention and action [42]. However, recovery is a long process that acts as a perfect opportunity to implement mindful interventions in order to rebuild and redevelop communities to make them more resilient and sustainable [2]. The information presented in this article can undoubtedly guide the public authorities to identify and implement various preventive and curative interventions that would be likely to reduce the extent and severity of the respiratory problems experienced by the victims. According to the lessons learned from this study and in the context of rapid and irremediable changes in socio-environmental conditions (due to climate change and increases in exposure and vulnerability factors), we must promote the recovery and resilience of affected communities through science-oriented interventions. This means that longitudinal surveys following floods that include recovery indicators need to be established through constant dedicated support from governmental and health authorities. This is especially crucial since more frequent and severe flooding events under climate change can trigger increased exposures and systemic risks [34]. The evolution of flood impacts needs to be evaluated over time to better protect the well-being and health of affected individuals.

## Figures and Tables

**Figure 1 ijerph-19-11738-f001:**
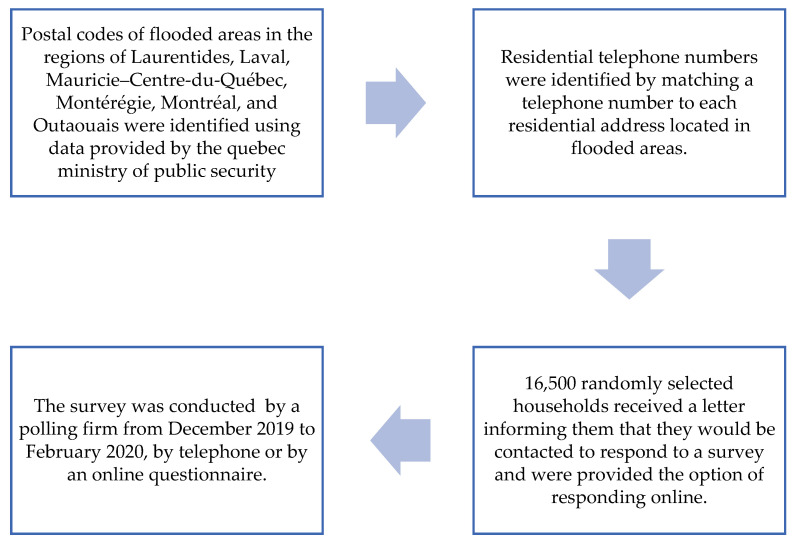
Methodology flow chart.

**Table 1 ijerph-19-11738-t001:** Flood exposure according to regions.

	Flooded	Disrupted	Non-Affected
Regions	*n* (%)	*n* (%)	*n* (%)
Laurentides (excluding Sainte-Marthe-sur-le-Lac)	42 (5.0)	209 (25.0)	585 (70.0)
Sainte-Marthe-sur-le-Lac	178 (30.3)	335 (57.1)	74 (12.6)
Laval	22 (16.4)	86 (64.2)	26 (19.4)
Outaouais	67 (8.4)	320 (40.3)	407 (51.3)
Mauricie-Centre du Québec	20 (3.1)	115 (17.9)	508 (79.0)
Montérégie	8 (2.5)	101 (31.0)	217 (66.6)
Montréal	12 (10.3)	64 (54.7)	41 (35.0)
All regions affected by floods	349 (10.2)	1230 (35.8)	1858 (54.1)

**Table 2 ijerph-19-11738-t002:** Prevalence of primary and secondary stressors according to regions affected by floods.

	Sainte-Marthe-sur-le-Lac	All Regions (Excluding Sainte-Marthe)	All Regions
	*n* (%)	*n* (%)	*n* (%)
Primary Stressors (Among All Respondents)			
Level of exposure			
Flooded	178 (30.3)	171 (6.0)	349 (10.2)
Disrupted	335 (57.1)	895 (31.4)	1230 (35.8)
Unaffected	74 (12.6)	1784 (62.6)	1858 (54.1)
Water level (at the first floor)			
No water	404 (69.3)	2617 (92.0)	3021 (88.2)
Less than 30 cm	36 (6.2)	129 (4.5)	165 (4.8)
30 to 100 cm	48 (8.2)	49 (1.7)	97 (2.8)
More than 100 cm	95 (16.3)	49 (1.7)	144 (4.2)
Extent of material losses			
No loss	369 (63.3)	2437 (86.1)	2806 (82.2)
Less than $25,000	52 (8.9)	304 (10.7)	356 (10.4)
25,000 to $49,999	63 (10.8)	39 (1.4)	102 (3.0)
$50,000 or more	99 (17.0)	51 (1.8)	150 (4.4)
Recurrence of floods			
Never flooded	355 (60.7)	2200 (77.6)	2555 (74.7)
Flooded in 2019 only	200 (34.2)	157 (5.5)	357 (10.4)
Flooded in 2019 and before 2019	30 (5.1)	479 (16.9)	14.9 (14.9)
Secondary stressors (among flooded or disrupted)			
Perception about concrete or moral help received			
More than expected	41 (18.8)	80 (22.1)	121 (20.9)
As much as expected	121 (55.5)	152 (42.0)	273 (47.1)
Less than expected	56 (25.7)	130 (35.9)	186 (32.1)
Financial sum received to meet costs			
All or most of the costs	78 (34.5)	125 (46.8)	203 (41.2)
About half the costs	78 (34.5)	71 (26.6)	149 (30.2)
Less than half the costs	70 (31.0)	71 (26.6)	141 (28.6)
Insurance covering floods			
Yes	245 (55.4)	478 (51.8)	723 (53.0)
No	197 (44.6)	444 (48.2)	641 (47.0)
Bank loan to meet expenses			
Yes	49 (16.9)	56 (8.5)	105 (11.1)
No	241 (83.1)	601 (91.5)	842 (88.9)
Normal reuse of all rooms			
Yes	426 (83.2)	993 (93.4)	1419 (90.1)
No	86 (16.8)	70 (6.6)	156 (9.9)

Note: Chi-square tests were performed to compare the prevalence of primary and secondary stressors in Sainte-Marthe-sur-le-Lac and the other participating regions.

**Table 3 ijerph-19-11738-t003:** Respiratory health according to flood exposure and region.

	Sainte-Marthe-sur-le-Lac		All Regions (Excluding Sainte-Marthe)		All Regions	
	ENT Symptoms	Respiratory Symptoms	ENT Symptoms	Respiratory Symptoms	ENT Symptoms	Respiratory Symptoms
	*n* (%)	*n* (%)	*n* (%)	*n* (%)	*n* (%)	*n* (%)
Flooded	61 (34.3) _a_	35 (19.7) _a_	60 (35.1) _a_	34 (19.9) _a_	121 (34.7) _a_	69 (19.8) _a_
Disrupted	64 (19.1) _b_	20 (6.0) _b_	203 (22.7) _b_	88 (9.8) _b_	267 (21.7) _b_	108 (8.8) _b_
Unaffected	13 (17.6) _b_	5 (6.8) _b_	259 (14.5) _c_	123 (6.9) _c_	272 (14.6) _c_	128 (6.9) _b_

Note 1: No significant differences between participating regions for the same exposure group (*p* ≥ 0.05) according to Chi-square test. Note 2: Exposure groups (observed for each type of symptom) with the same subscript letter have proportions that do not differ significantly from each other at the 0.05 level.

**Table 4 ijerph-19-11738-t004:** Respiratory health according to personal characteristics and flood exposure.

	ENT Symptoms			Respiratory Symptoms		
	Flooded	Disturbed	Unaffected	Flooded	Disturbed	Unaffected
	*n* (%)	*n* (%)	*n* (%)	*n* (%)	*n* (%)	*n* (%)
Sex						
Men	38 (30.2) _a_	85 (18.3) _a_	92 (11.9) _a_	22 (17.5) _a_	42 (9.1) _a_	46 (6.0) _a_
Women	83 (37.2) _a_	182 (23.8) _b_	180 (16.5) _b_	47 (21.1) _a_	66 (8.6) _a_	82 (7.5) _a_
Age						
18–44 years old	20 (32.2) _a_	43 (15.4) _a_	27 (8.8) _a_	12 (19.4) _a_	12 (4.3) _a_	11 (3.6) _a_
45–64 years old	62 (32.6) _a_	134 (23.2) _b_	98 (12.0) _a_	36 (18.9) _a_	59 (10.2) _b_	54 (6.6) _a,b_
More than 65 years old	39 (40.2) _a_	90 (24.1) _b_	147 (20.0) _b_	21 (21.6) _a_	37 (9.9) _b_	63 (8.6) _b_
Education						
High school or less	56 (37.1) _a_	116 (29.1) _a_	134 (16.0) _a_	31 (20.5) _a_	64 (16.1) _a_	77 (9.2) _a_
College	22 (27.5) _a_	64 (19.2) _b_	55 (12.6) _a_	14 (17.5) _a_	19 (5.7) _b_	25 (5.7) _b_
University	39 (35.1) _a_	85 (17.8) _b_	81 (14.4) _a_	23 (20.7) _a_	25 (5.2) _b_	25 (4.4) _b_
Annual household income						
Less than $29,999	17 (42.5) _a_	47 (31.8) _a_	79 (22.3) _a_	9 (22.5) _a_	27 (18.2) _a_	41 (11.5) _a_
30,000$ to $79,999	50 (35.2) _a_	108 (25.4) _a_	111 (16.1) _b_	32 (22.5) _a_	41 (9.6) _b_	45 (6.5) _b_
More than $80,000	33 (29.5) _a_	6 (15.5) _b_	53 (11.6) _c_	21 (18.8) _a_	22 (5.0) _c_	25 (5.5) _b_
Smoking status						
Yes	29 (39.2) _a_	51 (26.2) _a_	57 (20.7) _a_	21 (28.4) _a_	39 (20.0) _a_	47 (17.0) _a_
No	92 (33.6) _a_	216 (20.9) _a_	214 (13.6) _b_	48 (17.5) _b_	69 (6.7) _b_	81 (5.1) _b_

Note: Z-tests were performed for each type of flood exposure according to the socio-demographic factors. Subsets of socio-demographic factors with the same subscript letter have proportions that do not differ significantly from each other at the 0.05 level.

**Table 5 ijerph-19-11738-t005:** Associations between primary stressors and ENT and respiratory symptoms in all regions affected by flooding according to sex, age, education and smoking status.

	**ENT Symptoms**	**Crude Odds Ratios**		**Adjusted Odds Ratios**	
**Primary Stressors**	** *n* ** **(%)**	**OR**	**[95% CI]**	**OR**	**[95% CI]**
Level of exposure					
Flooded	121 (34.7)	**3.09**	**[2.40; 3.99]**	**3.18**	**[2.45; 4.14]**
Disturbed	267 (21.7)	**1.62**	**[1.34; 1.95]**	**1.76**	**[1.45; 2.14]**
Unaffected	272 (14.6)	1	Reference	1	Reference
Water Level (at first floor)					
No water	520 (17.2)	1	Reference	1	Reference
Less than 30 cm	49 (29.7)	**2.03**	**[1.44; 2.88]**	**1.96**	**[1.38; 2.79]**
30 to 100 cm	24 (24.7)	1.58	[1.00; 2.53]	1.54	[0.96; 2.49]
More than 100 cm	62 (43.1)	**3.64**	**[2.58; 5.13]**	**3.56**	**[2.51; 5.04]**
Extent of material losses					
No loss	453 (16.1)	1	Reference	1	Reference
Less than $25,000	108 (30.3)	**2.26**	**[1.77; 2.90]**	**2.15**	**[1.67; 2.77]**
25,000 to $49,999	30 (29.4)	**2.16**	**[1.40; 3.35]**	**2.06**	**[1.32; 3.20]**
More than $50,000 $	62 (41.3)	**3.66**	**[2.60; 5.14]**	**3.86**	**[2.72; 5.47]**
Recurrence of floods					
Never flooded	407 (15.9)	1	Reference	1	Reference
Flooded in 2019 only	407 (28.3)	**2.08**	**[1.62; 2.68]**	**2.05**	**[1.58; 2.65]**
Flooded in 2019 and before 2019	148 (29.1)	**2.16**	**[1.74; 2.69]**	**2.08**	**[1.67; 2.60]**
	**Respiratory Symptoms**	**Crude Odds Ratio**		**Adjusted Odds Ratio**	
	***n*** **(%)**	**OR**	**[IC 95%]**	**OR**	**[IC 95%]**
Level of exposure					
Flooded	69 (19.8)	**3.33**	**[2.42; 4.58]**	**3.41**	**[2.45; 4.75]**
Disturbed	108 (8.8)	1.30	[1.00; 1.70]	**1.45**	**[1.10; 1.91]**
Unaffected	128 (6.9)	1	Reference	1	Reference
Water Level (at first floor)					
No water	222 (7.3)	1	Reference	1	Reference
Less than 30 cm	30 (18.2)	**2.80**	**[1.84; 4.26]**	**2.58**	**[1.68; 3.98]**
30 to 100 cm	15 (15.5)	**2.31**	**[1.31; 4.07]**	**2.02**	**[1.13; 3.62]**
More than 100 cm	34 (23.6)	**3.90**	**[2.59; 5.86]**	**3.48**	**[2.28; 5.31]**
Extent of material losses					
No loss	198 (7.1)	1	Reference	1	Reference
Less than $25,000	53 (14.9)	**2.30**	**[1.66; 3.19]**	**2.08**	**[1.49; 2.91]**
25,000 to $49,999	19 (18.6)	**3.02**	**[1.79; 5.07]**	**2.67**	**[1.57; 4.54]**
More than $50,000	30 (20.0)	**3.29**	**[2.15; 5.04]**	**3.64**	**[2.34; 5.66]**
Recurrence of floods					
Never flooded	179 (7.0)	1	Reference	1	Reference
Flooded in 2019 only	61 (17.1)	**2.74**	**[2.00; 3.75]**	**2.74**	**[1.98; 3.78]**
Flooded in 2019 and before 2019	64 (12.6)	**1.91**	**[1.41; 2.58]**	**1.79**	**[1.31; 2.44]**

Note: Odds ratios in bold are statistically significant *p* ≤ 0.05.

**Table 6 ijerph-19-11738-t006:** Associations between secondary stressors and ENT and respiratory symptoms in all regions affected by flooding according to personal characteristics (sex, age, education and smoking status).

	**ENT Symptoms**	**Crude Odds Ratios**		**Adjusted Odds Ratios**	
**Secondary Stressors (Among Flooded or Disrupted)**	** *n* ** **(%)**	**RC**	**[IC 95%]**	**RC**	**[IC 95%]**
Level of satisfaction about concrete or moral help received					
More than expected	31 (25.6)	1	Reference	1	Reference
As much as expected	81 (29.7)	1.23	[0.76; 1.99]	1.23	[0.75; 2.01]
Less than expected	70 (37.6)	**1.75**	**[1.06; 2.90]**	**1.87**	**[1.11; 3.13]**
Financial sum received to cover damages					
All or most of the costs	50 (24.6)	1	Reference	1	Reference
About half the cost	46 (30.9)	1.37	[0.85; 2.19]	1.42	[0.88; 2.30]
Less than half the costs	60 (42.6)	**2.27**	**[1.43; 3.60]**	**2.44**	**[1.51; 3.93]**
Insurance covering floods					
Yes	172 (23.8)	1	Reference	1	Reference
No	173 (27.0)	1.18	[0.93; 1.51]	1.15	[0.90; 1.48]
Bank loan to meet expenses					
Yes	48 (45.7)	**2.34**	**[1.55; 3.53]**	**2.54**	**[1.66; 3.89]**
No	223 (26.5)	1	Reference	1	Reference
Normal reuse of all rooms					
Yes	316 (22.3)	1	Reference	1	Reference
No	70 (44.9)	**2.84**	**[2.02; 3.99]**	**2.77**	**[1.96; 3.91]**
	**Respiratory symptoms**	**Crude Odds Ratios**		**Adjusted Odds Ratios**	
**Secondary Stressors (among Flooded or Disrupted)**	** *n* ** **(%)**	**OR**	**[IC 95%]**	**OR**	**[IC 95%]**
Level of satisfaction about Concrete or moral help received					
More than expected	15 (12.5)	1	Reference	1	Reference
As much as expected	38 (13.9)	1.14	[0.60; 2.17]	1.10	[0.58; 2.01]
Less than expected	35 (18.8)	1.64	[0.85; 3.15]	1.61	[0.82; 3.13]
Financial sum received to cover damages					
All or most of the costs	25 (12.3)	1	Reference	1	Reference
About half the costs	21 (14.1)	1.17	[0.63; 2.18]	1.15	[0.61; 2.17]
Less than half the costs	33 (23.4)	**2.18**	**[1.23; 3.86]**	**2.20**	**[1.24; 3.98]**
Insurance covering floods					
Yes	67 (9.3)	1	Reference	1	Reference
No	94 (14.7)	**1.68**	**[1.20; 2.35]**	**1.56**	**[1.12; 2.21]**
Bank loan to meet expenses					
Yes	26 (24.8)	**2.24**	**[1.37; 3.64]**	**2.48**	**[1.50; 4.10]**
No	108 (12.8)	1	Reference	1	Reference
Normal reuse of all rooms					
Yes	134 (9.4)	1	Reference	1	Reference
No	41 (26.3)	**3.42**	**[2.30; 5.09]**	**3.41**	**[2.27; 5.13]**

Note: Odds ratios in bold are statistically significant *p* ≤ 0.05.

## Data Availability

The data presented in this study are available on request from the corresponding author. The data are not publicly available due to participant confidentiality.

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
