# Peer review of "Respiratory and Otolaryngology Symptoms Following the 2019 Spring Floods in Quebec"

_ijerph, 2022, doi:10.3390/ijerph191811738_

Round 1

Reviewer 1 Report

Thank you for submitting your manuscript to the International Journal of
Environmental Research and Public Health. Generally, the topic fits into the
scope of the journal, and the structure respects Scientific Best Practice.
However, the content requires revision. In the literature review, it is important that the scientific novelty of the work is established through a critical analysis of related literature. With this, followng questions must be clarified: How does the present work contribute towards the gaps identified? How does it improve upon previous work? It is recommended that a short discussion of the novel contribution of each reference cited shall be provided to give readers a better understanding of their relevance.
Putting the scientific motivation more clear will also help you to
identify the novelties that characterises a scientific publication. The methodology should be improved. I strongly recommend to include a flow chart illustrating the steps of the methodology in the beginning of the methodology section. Moreover, the questionnaire(s) need to be
included as supplementary materials. The statistical methods applied need
to be described. In the results section it should be explained how the correlation of the
health data with flood events was made. In the conclusions, in addition to summarising the actions taken and results, please strengthen the explanation of their significance. It is recommended to use quantitative reasoning comparing with appropriate benchmarks, especially those stemming from previous work.  

Reviewer 2 Report

The title referred to respiratory health but included a lot of ENT systems.

Could be cited also the article of Saulnier and col. The effect of seasonal floods on health: Analisis of six years of National health data and floods maps (2018

The questionnaires were validated?

The results in the tables must be clear. (+) or (-) maybe will be better to be expressed as we are familiar with the P value.  

The row between 49 and 92 should be moved during the discussion

Round 2

Reviewer 1 Report

Thank you for the submission of the revision. My comments have been considered.